# The theoretical understanding of risk perception and dual-method contraceptive decision-making among Texas adolescent and young adult cisgender females

Lauren Holt[1]*, Amy Corneli[2], Eleanor Stevenson[1], Jonas J. Swartz[3], Michael V. Relf[1]

1 School of Nursing, Duke University, Durham, North Carolina, United States of America, 2 Department of Population Health Sciences, Duke University, Durham, North Carolina, United States of America, 3 Department of Obstetrics & Gynecology, Duke University Medical Center, Durham, North Carolina, United States of America

* Lauren.holt427@gmail.com

## Abstract

In this paper we use a constructivist grounded theory approach to conduct individual, qualitative interviews with a racially and ethnically diverse group of adolescent and young adult cisgender females (AYACF) (18–24 years of age) to generate theoretical knowledge of the relationship between Social Determinants of Health (SDOH), STI and HIV risk perception, and dual-method contraceptive decision-making (the use of, both, a prescribed contraceptive [commonly referred to as "birth control"] and a condom during vaginal intercourse). Twenty-five AYACF were selected through theoretical sampling. Data were collected using semi-structured interviews and analyzed using constant comparison. AYACF expressed having a greater fear of pregnancy than sexually transmitted infections (STIs). They described deferring to their partners and making assumptions about their partner's STI status based on partner and relationship factors, therefore not needing to engage in condom negotiation. Mother-daughter sexual health conversations and state school-based sex education and abortion policies influenced participants' STI risk perception and dual-method contraceptive decision-making. Participants who were diagnosed with an STI reported experiencing a shift in perception about their likelihood of acquiring an STI, often resulting in behavior change. These findings are important as they provide insight into the complex decision-making processes, influenced by SDOH, that a racially and ethnically diverse group of AYACF associate with STI risk perception and dual-method contraceptive use.

## Introduction

Sexually transmitted infections (STIs), including HIV, are a significant public health concern. In Texas, STIs are on the rise as primary syphilis rates doubled over the last

**Data availability statement:** Data cannot be shared publicly or with other researchers outside of those involved with the study due to the sensitive nature of the topic and the need to maintain participant privacy.Further, it was not indicated to participants that the

de-identified data would be made publicly available at the conclusion of the study. As such, because the participants were not made aware that this would happen, the data cannot be made publicly available at this time as imposed by Duke IRB. Data requests can be made to Duke IRB Suite 900 Erwin Square 2200 West Main Street Campus Box # 104026 Durham, NC 27705 Phone: (919) 668-5111 Email:ResearchServiceDesk@duke.edu.

**Funding:** LH -Grant number: 383000723 (Sigma Theta Tau/Association of Nurses in AIDS Care) -Funders: Sigma Theta Tau/ Association of Nurses in AIDS Care; Margolis Scholars Program -Website for Sigma: https://www.sigmanursing.org/advance-el-evate/research/research-grants/sigma-the-ta-tau-international-association-of-nurs-es-in-aids-care-grant -Website for Margolis: https://healthpolicy.duke.edu/scholars#:~:-text=The%20Margolis%20Scholars%20program%20is%0a%20a%20selective%20program,well%20as%20leadership%20poten-tial%20to%20improve%20health%20policy. -No, the funders did not play any role in the study design, data collection and analysis, decision to publish, or preparation of the manuscript.

**Competing interests:** The authors have declared that no competing interests exist.

five years, and congenital syphilis rates are at an all-time high [1]. Adolescent and young adult cisgender females (AYACF) between 18 and 24 years old have experienced a consistent increase in syphilis over the last five years, while accounting for higher rates of chlamydia than any age or gender group [1]. Further, AYACF belonging to structurally oppressed racial and ethnic groups are disproportionately more affected by STIs and HIV than White AYACF; Black females are five times more likely to be diagnosed with chlamydia and three times more likely to be diagnosed with HIV, while Hispanic females are twice as likely to be diagnosed with syphilis [2]. Moreover, American Indian/Alaskan Native females are six times more likely to be diagnosed with gonorrhea, and five times more likely to be diagnosed with syphilis than their White counterparts [2].

Despite rising STI rates, pregnancy rates have steadily declined among Texas AYACF [3], a feat largely attributed to an increase in access and utilization of prescribed contraceptive methods (i.e., intrauterine devices (IUDs), birth control pills, hormonal implants, injections, patches, and rings) [3–5]. Although prescribed contraceptives are 90–99% effective in reducing the risk of pregnancy when used correctly [6], they do not protect against STIs or HIV.

Unlike prescribed contraceptives, condoms can simultaneously prevent pregnancy, STIs and HIV, however, they have a higher risk of user error than prescribed contraceptives [7]. When used correctly, condoms are 98% effective, however user error such as breakage, slippage, improper application, or structural damage from improper storage can decrease their effectiveness to 87% [7]. Further, condoms need to be used consistently to prevent pregnancy, HIV, and STDs, and only 23% of AYACF report using condoms during every sexual encounter [8] Due to their low utilization rates and high risk for user error, the American College of Gynecologists recommends that sexually active AYACF use a prescribed contraceptive method as their primary source of pregnancy prevention in conjunction with a condom to reduce STI and HIV risk during sexual intercourse [16–18]. The use of, both, a prescribed contraceptive and a condom during sexual intercourse is known as dual-method contraceptive use [9], and will be defined as such throughout this paper.

AYACF who use prescribed contraceptives are less likely to practice dual-method contraceptive use [10,11], and adolescent females who use long-acting reversible contraceptives (LARCs) such as IUDs and implants are less likely to practice dual-method contraceptive use and have more sexual partners than those using short-acting reversible contraceptives (SARCs) such as oral birth control pills [12–16]. Professional associations and practice guidelines promote LARCs as the first-line contraceptive method among adolescent women due to they're high efficacy rate (>99%) [17–19]. Additionally, recently enacted laws restricting abortion access has contributed to an increase in LARC use [20]. The increase in LARC promotion and use highlights the need to investigate contributing factors to dual-method contraceptive decision-making among AYACF.

Risk perception analysis is an important factor in dual-method contraceptive decision-making as AYACF with a high STI and HIV risk perception are more likely to initiate dual-method contraceptive use than those who perceive themselves as

low risk [13,21–23]. While risk perception analysis has been identified as an integral part of decision-making [24], the processes AYACF associate with STI and HIV risk perception analysis and dual-method contraceptive decision-making among AYACF are poorly understood.

The overall objective of this qualitative study was to examine the factors adolescent and young adult cisgender females (AYACF) (18–24 years of age) consider when deciding to engage in dual-method contraceptive use. Using a constructivist grounded theory approach, we conducted individual, qualitative semi-structured interviews with a racialy and ethnically diverse group of sexually active AYACF using prescribed contraceptives who have had vaginal intercourse in the last 12 months to generate theoretical knowledge of the relationship between Social Determinants of Health (SDOH), STI and HIV risk perception, and dual-method contraceptive decision-making to inform the development of a conceptual model for future theoretical testing and intervention development.

## Background: Social determinants of health

The World Health Organization (WHO) defines SDOH as the circumstances under which individuals are born, raised, live, work, and age, and the available systems to deal with illness [25]. These circumstances are shaped by structural determinants which encompass social and political mechanisms that influence individual and collective access to resources, further contributing to social class division [26]. Structural determinants such as statewide policies pertaining to sexual and reproductive health access and education, and societal and cultural norms and values influence dual-method contraceptive use among AYACF, resulting in STI and HIV disparities [27–29]. For example, only 18 states require sex education content to be medically accurate and only 27 states require HIV education to be included in sex education curricula [30]. Further, only 20 states are required to provide students with information on contraceptive methods [30]. As a result, only 52% of U.S. youth receive education about where to obtain prescribed contraceptive methods and only 59% are educated on condom application [27].

Additionally, abortion policy has influenced contraceptive decision-making, as the overturn of *Roe V. Wade* in June 2022 prompted re-evaluation of pregnancy prevention for many. For example, since 2022, LARC use has increased, and the number of tubal sterilization and vasectomy procedures increased among young adults ages 18–30 years old [20,31]. Moreover, the use of emergency contraception (commonly referred to as Plan B) surged after June 2022 (and has since subsided) while oral contraceptives have decreased in states where abortion laws are the most restrictive [32].

Societal and cultural norms and values are structural determinants which determine STI and HIV risk and can inadvertently serve as a barrier to engaging in dual-method contraceptive use, if not acknowledged [28,33,34]. Gendered societal expectations (also called "gender roles") consist of how society expects women and men to behave, express themselves, and interact with others [35]. For example, in most cultures women are expected and perceived to be selfless, polite, and submissive, while men are expected to be strong, assertive, and dominant [35]. As a result of gendered societal expectations, uneven relationship dynamics and are likely to occur, resulting in women feeling less empowered to negotiate condom use [28,29,36,37].

In the context of SDOH, structural determinants operate through intermediary determinants which directly affect dual-method contraceptive use among AYACF [26]. Health care access and quality play a critical role in determining dual-method contraceptive use as AYACF with access to health insurance are more likely to engage in dual-method contraceptive use and less likely to report an STI and HIV than those who are medically uninsured [38,39].

Additionally, psychosocial factors such as the interpersonal relationship dynamics between an AYACF and her partner determine dual-method contraceptive use as AYACF who report higher levels of relationship trust and commitment are less likely to practice dual-method contraceptive use than those in less trusting or committed relationships [13,40]. Meanwhile, negative relationship factors such as violence and conflict are associated with inconsistent, or lack of, dual-method contraceptive use [41,42]. To understand the processes influencing with dual-method contraceptive decision-making and

STI and HIV risk perception among AYACF, healthcare access, and quality and relationship factors must be taken into consideration.

## Methods

### Study design

This study implemented a constructivist grounded theory qualitative design to generate a middle-range theory related to STI and HIV risk perception analysis and dual-method contraception decision-making among sexually active AYACF [43]. The proposed study integrated the essential components by 1) theoretically sampling participants to further explore emerging data, 2) conducting in-depth interviews, 3) developing and using memos to ensure dependability and confirmability in increasing the trustworthiness of the data and findings and, 4) using a constant comparison approach in data analysis [43]. The primary data collection strategy was individual semi-structured qualitative interviews collected at one point in time to provide an in-depth exploration of the participants' experiences and situations [43].

Unlike Classic or Straussean grounded theory approaches, constructivist grounded theory acknowledges that the researcher is not neutral in their observations or a value-free expert, requiring that the researcher address how their own privileges and preconceptions may shape the data analyses [43]. A constructivist approach was most suitable for this study as the author's own experience as an AYACF with prior academic knowledge of prescribed contraceptive use needed to be taken into consideration throughout data collection and analysis.

### Participants and setting

In conducting a constructivist grounded theory study, it was imperative to select participants who had first-hand experience that would address the aims of the proposed study [43]. To understand the processes of STI and HIV risk perception and dual-method contraceptive decision-making among sexually active AYACF using prescribed contraceptives, participants must have met the following inclusion criteria: 1) self-identify as a cisgender female; 2) be 18–24 years old; 3) report vaginal intercourse in the last 12 months; and 4) currently using a prescribed contraceptive method such as LARCs (i.e. IUD's, implants) or SARCs (i.e., contraceptive pills, injections, patches, rings).

Study recruitment occurred from January 11, 2023 to November 30, 2023 at a health clinic in urban Texas that provides free gynecological, birth control, family planning, and STI screening and treatment services to any adolescent and young adult residing in one of the largest and most racially and ethnically diverse counties in the US [44,45]. The racial demographics for the county where recruitment took place is Hispanic/Latino (44%), White/Non-Hispanic (23.6%), Black/African-American (22.5%), Asian (6.8%), Other (1%). Socioeconomically, the median household income is $71,811 with 13.6% of families living below the poverty line. Politically, the county leans democratic in a conservative state. The clinic is located in the middle of two major districts within the county. One district is predominately African-American (36%) with a median household income of $65,052 [46], while the other district is predominately Hispanic (91%) with a median household income of $52,611 [47].Further, the county where the health clinic resides reports the highest STI case numbers in the state of Texas while accounting for 1 in 4 new HIV diagnoses at a rate that is almost twice the state average [48].

### Description of data collected

A semi-structured interview format was appropriate for this study as it facilitated the exploration of researcher-initiated topics, while still allowing participants to freely discuss thoughts or ideas that might not have been considered by the research team [49]. Interview questions were drafted by the research team who specialize in sexual and reproductive health to ensure content validity. Further, participant input contributed to the development of additional questions that were asked in future interviews. After the fourteenth interview, triangulation was used to identify new questions to add to the interview guide that would elicit further discussion about perceived monogamy and STI risk. While searching for qualitative

studies that focused on condom use among adolescents and young adults, the research team identified a study by Bolton, McKay [50] and adapted questions from their interview guide starting with the fifteenth interview. Based on observations from the first fourteen interviews, the research team added questions such as "How did you know you were/are in a monogamous relationship?"," What was happening in the relationship at the time?" and "What do you think is the likelihood that you will acquire an STI?" [50]. Key questions that guided the interviews are presented in Table 1. The addition of these questions provided further insight into how the participants perceived their partners and themselves while in a romantic and/or sexual relationship, guiding their contraceptive decision-making.

Before the interview, each participant completed a demographic data form which included age, sex, gender identity, race and ethnicity information, relationship status, and type of prescribed contraception in use.

## Study procedures

Theoretical sampling was implemented based on age, race, ethnicity, and type of prescribed contraceptive (SARC vs. LARC) [43]. A total of twenty-five participants were interviewed for this study. Fifteen of the selected participants were recruited in person at the health clinic and ten participants were referred to the research team by other participants. Once participants agreed to the interview, they were provided with a consent form to be signed before the interview. Participants were given the option to be interviewed in a private room at the clinic in-person or virtually in a private location of their choice. A total of twenty-five one-time, semi-structured interviews were conducted via Zoom after obtaining informed consent. All twenty-five interviews were conducted by one member of the research team (LH) who has training in qualitative research methods. Following theoretical sampling procedures [43], after one interview was completed, transcribed, and analyzed, the next participant was theoretically recruited and selected for an interview to ensure the representativeness of cisgender adolescent young adult women's experiences. Characteristics of previous participants (i.e., age, race, ethnicity, education, birth control method) were components considered in the theoretical sampling process. This sampling approach helped to ensure the confirmability of diverse experiences.

Individual semi-structured interviews took place virtually via Zoom and were recorded with an encrypted audio recording device. All interviews were transcribed verbatim by a transcription service and checked against the recording for accuracy. Investigator memos were written immediately following the interviews and supplemented throughout the data analysis process to capture thoughts and develop connections [43]. For participating in the study, each participant was compensated with a $40 Amazon gift card.

## Data analysis

An inductive qualitative content analysis, using constant comparison, was conducted to explain the processes AYACF consider with risk perception analysis and decision-making related to dual-method contraceptive use among sexually active AYACF using prescribed contraceptives [43]. The transcribed verbatim transcripts were imported into NVivo for coding and analysis [51]. Constant comparison analysis was performed between two members of the research team who have experience in qualitative data analysis and sexual and reproductive health research throughout the length of the data collection period and during three phases of coding- initial, focused, and theoretical.

First, initial coding was performed to ensure fit and relevance of the analyzed data and to identify connections between the participant's actions and larger social processes to help to identify the possible paths that the data might take [43]. The initial coding process began shortly after an interview took place and consisted of line-by-line coding of the transcribed interviews, field notes, and memos in order to highlight recurring words and immediately compare and contrast findings from previous interviews [52]. Next, the focused coding process occurred by assessing and comparing the initial codes to distinguish which codes had the greatest significance and theoretical direction [43]. To address coding discrepancies, analysts discussed their rationale and reached a consensus on the most appropriate coding approach. When consensus could not be achieved, members of the research team with expertise in sexual and reproductive health and/or qualitative

**Table 1. Social Determinants of Health Interview Guide.**

Rapport-building questions:
• Tell me a little about yourself.
• What interested you in this study?

*Structural Determinants*
STI/HIV Prevention *(Sex Education Policy/Societal and Cultural Norms and Values)*
Question 1: As previously mentioned in the screening, this is a study for sexually active women who are using prescribed contraceptives, commonly referred to as birth control, and we are interested in learning how they perceive STI/HIV risk. What do you think are the most important ways a young woman can protect herself against pregnancy and sexually transmitted infections (STIs) including HIV?
• Probe: Tell me about your experience in learning about prescribed contraceptive and condom use.
• Probe: Tell me about your sexual health education experience in school
• Probe: Can you share with me how women in your community protect themselves from pregnancy and STIs, including HIV.
• *Probe: Tell me about how one's culture might influence using a birth control and/or condom during sex*
• Probe: How might where someone grew up influence their understanding of sexual health and pre-scribed birth control

*Intermediary Determinants*
Prescribed Contraceptives *(Health System and Psychosocial)*
Question 2: Could you tell me about the events that led to your decision to seek prescribed contraceptive?
• Probe: As you thought about seeking prescribed contraceptives, who were the individuals you talked to for guidance?
• Probe: Can you share with me the information you were considering before starting a prescribed contraceptive?
• Probe: Tell me about any barriers you faced in obtaining your prescribed contraceptive?
• Probe: Tell me about your interaction with your provider during the visit you were prescribed contraceptive
• Probe: Tell me about the things you were thinking about when you made the decision to start a specific prescribed contraceptive.
Prescribed Contraceptive and Condom Use *(Psychosocial)*
Question 3: How would you describe how you viewed condom use before you got on a prescribed contraceptive?
• Probe: What pregnancy prevention methods did you use before starting a prescribed contraceptive?
• Probe: What contributed to your decision to use (or not to use) condoms before starting a prescribed contraceptive?
• Probe: Could you tell me about how your views towards condom use may have changed since starting a prescribed contraceptive?
Sexual Partners *(Psychosocial)*
Question 4: Since we have been discussing condom use, I would like to learn about your current or most recent sexual partner.
• Probe: Can you share with me conversations that you and your partner had regarding sexually transmit-ted infection (STI) prevention?
• Probe: Can you tell me how you both came to this understanding?
For participants who are currently in or had previously been in a monogamous relationship:
• *How did you know you were/are in a monogamous relationship? [50]
• *Probe: What was happening in the relationship at that time? [50]
For all participants:
• *What do you think is the likelihood that you will acquire an STI? [50]
Closing: Is there anything else you would like to share with me that we haven't talked about today?

• Before we end our conversation, I wanted to give you an opportunity to ask any questions.
*Questions inserted after fourteenth interview

research were consulted. Once the focused coding of the transcribed interview, field notes, and memos were complete, we performed theoretical coding to conceptualize how the focused codes were related to specific categories [43].

In order to add precision and clarity to our data analysis, theoretical coding was conducted by 1) analyzing and synthe-sizing relationships between focused codes and 2) building a conceptual model explaining the processes AYACF consider

with risk perception analysis and decision-making related to dual-method contraceptive use among sexually active AYACF using prescribed contraceptives [43,53]. The development of the conceptual model began during the theoretical coding phase of the first interview and occurred throughout the theoretical coding phase of the remaining interviews. By the thirteenth interview, major themes began to emerge, and were solidified by the twentieth interview. By the twenty-fifth interview data saturation had been achieved as no new themes had emerged and sufficient data was present to support the existing themes within the conceptual model.

### Trustworthiness and rigor

To ensure credibility, triangulation was used to compare study findings to the literature throughout the data analysis process. Over the course of the study, data, procedures, and tools were constantly compared against the literature to assess the need for alterations. During the data collection and analysis process, emerging themes were compared to other study findings to assess the need to develop new interview questions to be added to the interview guide. To ensure confirmability, reflexive journaling was conducted after each interview to capture thoughts and perceptions and identify any potential biases. Since the individual conducting the interviews (LH) once identified as an AYACF, reflexive journaling was essential in identifying and processing any potential biases that existed based on prior experiences. To ensure dependability, an audit trail detailing the developmental and implementation stages of this study was kept to verify the accuracy and reliability of the data collected.

### Ethical considerations

Approval to conduct this study was obtained from the Duke Institutional Review Board (Pro00111825). While participant risk in this study was minimal, measures were taken to ensure the physical, psychological and emotional well-being of participants were addressed. The sensitive nature of the topic had the potential to reveal incidents of sexual abuse and violence. Thus, measures were in place to report to legal authorities and provide mental health resources had a participant shared an experience of sexual abuse or violence.

Further, to minimize the potential for emotional distress during interviews, participants were encouraged to only share information for which they were comfortable disclosing. If questions or responses lead to emotional distress or discomfort, each participant was provided with an opportunity to pause, reschedule, or stop the interview. To build trust and ensure all participants felt respected in an interview setting, the interviewer (LH) attended and participated in a qualitative research course specific to interviewing participants belonging to structurally oppressed racial and ethnic groups.

The threat to confidentiality was reduced by using a pseudonym or participant code number in place of the participants' names. No actual names of participants were used in the study write-up. During data collection, paper-based sources of data and audio recordings were kept in a secured filing cabinet in a locked office. De-identified electronic data, including transcribed interviews, demographic information and field notes were password-protected and stored in an encrypted database on a secure Duke University server. Access to both paper-based and electronic databases were limited to the LH (interviewer and coder) and MR (coder).

## Results

As described in Table 2, of the twenty-five AYACF recruited for this study, the mean age of participants was 20.7 years and represented diverse racial (African American, 44%; White, 4%; Asian, 4%; Other, 16%) and ethnic identities (Hispanic, 48%; Non-Hispanic, 48%; Unknown, 2%). The participants were mainly college-educated (80%) without children (88%) and currently in a monogamous relationship with one male partner (52%). Additionally, most participants had previously been diagnosed with an STI (52%) and were currently using Depo Provera (40%) as their prescribed contraceptive method. Interviews ranged from 18 to 55 minutes, with an average of 38 minutes per interview.

**Table 2. Participant Demographics.**

| Age | # | % |
|---|---|---|
| 18 years old | 2 | 8% |
| 19 years old | 3 | 12% |
| 20 years old | 11 | 44% |
| 21 years old | 2 | 8% |
| 22 years old | 2 | 8% |
| 23 years old | 1 | 4% |
| 24 years old | 4 | 16% |
| **Race/Ethnicity** | **#** | **%** |
| White | 1 | 4% |
| Black | 11 | 44% |
| Hispanic | 8 | 32% |
| Asian | 1 | 4% |
| Other | 4 | 16% |
| **Education** | **#** | **%** |
| Current College Student | 19 | 76% |
| Graduated College | 1 | 4% |
| Non-College | 4 | 16% |
| High School | 1 | 4% |
| **Previous STI Diagnosis** | **#** | **%** |
| Yes | 13 | 52% |
| No | 12 | 48% |
| **Birth Control Method** | **#** | **%** |
| Shot | 10 | 40% |
| Implant | 5 | 20% |
| Pill | 8 | 32% |
| Ring | 1 | 4% |
| Patch | 1 | 4% |
| **Relationship Status** | **#** | **%** |
| Single and having casual sex with one male partner | 8 | 32% |
| Single and having casual sex with more than one male partner | 4 | 16% |
| In monogamous relationship with one male partner | 13 | 52% |
| In open relationship with more than one male partner | 0 | 0% |

The data analysis revealed that the process of STI risk perception analysis and dual-method contraceptive decision-making consists of the following overarching themes 1) Fear of pregnancy greater than fear of STIs, 2) Making assumptions and partner deferral, and 3) STI Diagnosis: A new reality as seen in Fig 1.

For AYACF, SDOH and personal background perpetuate the belief that pregnancy is a more significant concern than STIs. As AYACF navigate sexual relationships, they make assumptions about a sexual partner's STI status based on specific partner characteristics and relationship dynamics. Together, relationship dynamics and partner characteristics influence the decision to defer to the sexual partner for dual-method contraceptive decision-making. Additionally, a general lack of concern for STIs contributed to the decision to forgo condom use and rely on STI testing after engaging in sexual intercourse without a condom. As a result of deferring to the sexual partner and substituting condom use for STI testing, an STI diagnosis occurred, causing a shift in STI risk perception, which at times resulted in a behavior change.

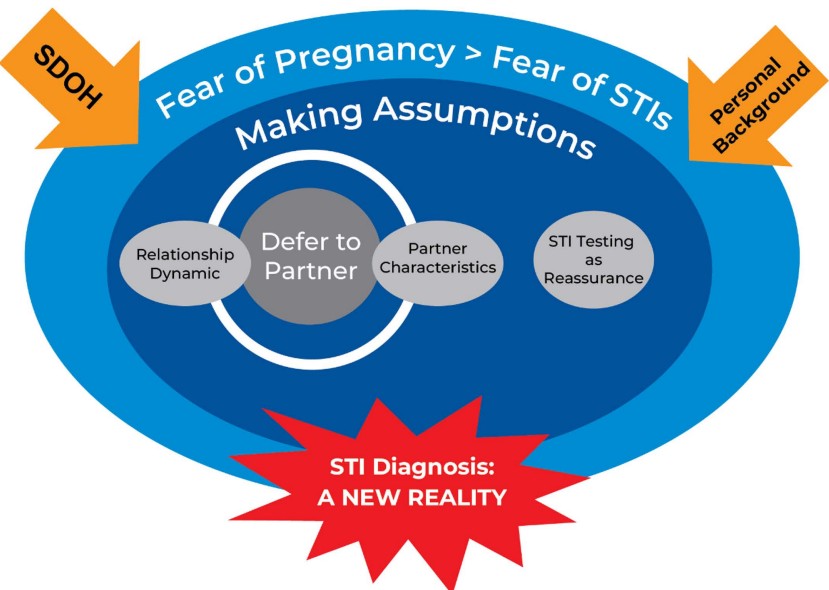

**Fig 1. Dual-method contraception decision-making factors.**

### Fear of pregnancy > fear of STIs

For many participants, the fear of an unwanted pregnancy was far greater than their fear of acquiring an STI. Although all participants were prescribed contraceptive users, many reported feeling *scared* and *anxious* about getting pregnant. For example, a twenty-one-year-old Asian college student compared her fear of pregnancy to a feeling of *impending doom*. While the fear of pregnancy was palpable among participants, their fear of STIs was minimal, playing a significant role in their STI risk perception. A twenty-year-old Hispanic college student shared her perspective on pregnancy and STIs:

> *Well, I think pregnancy is definitely a lifetime, like a for-life commitment. And that's definitely way scarier, even though STIs are scary for sure. I don't know. I just feel like when you think of pregnancy, it's like really scary. Like, really scary, I mean, especially for women. I've always just been scared of my life stopping there.*

Having an overwhelming fear of unwanted pregnancy contributed to dual-method contraceptive decision-making as several participants stopped using condoms once initiating a prescribed contraceptive. For example, a twenty-year-old Hispanic college student with a previous STI diagnosis described her experience of using condoms before initiating pre-scribed contraceptive use:

> *So I have used a condom since, like, I started the first time* [being sexually active]. *And I've been using a condom since then until I went on birth control for the whole time, and then I stopped using a condom.*

Further, participants who engaged in dual-method contraceptive use reported using condoms for additional pregnancy protection as opposed to STI and HIV protection. A twenty-four-year-old Hispanic college student explained why she used condoms in addition to a prescribed contraceptive.

> *I think it was mostly, like, my thinking of condoms was, like, me just not getting pregnant. It was, like, another barrier for birth control, basically.*

## Social determinants of health

Recent changes to Texas abortion laws exacerbated participants' fear of unwanted pregnancy. As a result of newly implemented state laws banning abortion, participants considered additional ways to prevent pregnancy and detect pregnancy at the earliest stage possible. In the quote below, a nineteen-year-old African American college student shared her experience:

*I feel like this makes me more conscious of knowing, like, I need to know right now if I think I'm pregnant because we should not be wasting time.*

Further, participants contemplated how they could access or self-induce an abortion in the instance that they became pregnant. For example, a nineteen-year-old college student who identified as Hispanic, African American, and White shared how she would approach an unplanned pregnancy.

*I'm pretty confident that I would catch it* [pregnancy] *in time, and I will just get mailed in some abortion pills and just take that and naturally miscarry.*

A twenty-one-year-old Asian college student shared a similar perspective on what she would do if she were to experience an unplanned pregnancy.

*I can't afford an abortion. Do I have to buy a bunch of vitamin C? To have like one of those, like self-abortions or whatever I hear people talking about like a hanger.*

School-based sex education also played a significant role in how participants perceived pregnancy and STIs. For example, a twenty-one-year-old Asian college student shared her sex education experience:

*They just instill a lot of fear of not getting pregnant at an early age and not getting STDs. And it's, it's effective because I'm deadly scared of getting pregnant at my age.*

The majority of participants received a sexual health education in school that primarily focused on abstinence. A nineteen-year-old African American college student with a prior STI diagnosis said:

*I'm not gonna lie, like when I was younger and like Sex-Ed and stuff, they didn't really like teach it. They taught me more like abstain. Because they're like the best way to prevent everything, just abstain.*

## Personal background

*Breaking the cycle* of becoming a mother at a young age was especially important to participants whose mothers had experienced an unplanned pregnancy in their adolescence or early twenties. The hardship experienced by their mothers navigating young motherhood further exacerbated participants' fear of having an unplanned pregnancy. A twenty-year-old Hispanic college student shared how her mother's experience influenced her decision to be on a prescribed contraceptive.

*My mom's experiences have also had a big impact on how breaking the cycle is actually important and how it's up to you whether you want to be better for yourself and for your family.*

Similarly, a twenty-year-old Hispanic college student shared how her mother's openness about her own life experiences shaped her contraceptive decision-making.

*…my mom is very open to her past life or her childhood, and she talks about how those experiences have like made her feel, and it just kind of like impacts me because…I don't want to like I guess disappoint her in the way where it's like I'm gonna do the same or like all her sacrifices are for nothing.*

Relationships between participants and their mothers played a major role in how participants perceived STI risk. For example, participants who did not talk to their mothers about sex-related topics such as pregnancy and STI prevention methods were less likely to practice dual-method contraceptive use and more likely to have had a previous STI diagnosis compared to participants who had discussed sex-related topics with their mother. Below, a nineteen-year-old college student with a prior STI diagnosis who identified as African-American, Hispanic, and White shared her experience of attempting to discuss sex and pregnancy prevention with her mother.

*And my mom she just kind of was always kind of like, was kind of shushing me. She was like, oh no, like don't have sex like, I don't want to, I don't want to know about that, you have to be a virgin. Like she, she wanted, I think she would prefer if I was a virgin until I'm like 20, 23 or something.*

**Making assumptions: Participant characteristics, relationship dynamic, and defer to partner**

Instead of using condoms or inquiring about condom use, participants made assumptions about their partner's STI status based on specific partner characteristics. For example, age played a significant role in STI risk perception analysis and dual-method contraceptive decision-making as participants with older sexual partners were more likely to assume that their partner did not have an STI, resulting in less dual-method contraceptive use. Older partners were *mature* and *responsible*. However, they rarely used condoms or had conversations about condoms or STIs before engaging in sex with the participant. In the quote below, a twenty-two-year-old Hispanic college student recalled a conversation she had with a friend regarding her older sexual partner.

*I had mentioned to her* [friend] *that I was seeing this older guy. He was 21 at the time. I was like, oh, I really liked him, like, we already have sex, but he tends to pull out. I was like, do you think that's okay?*

Additionally, participants with older partners were more likely to defer to their partners for condom use decision-making. Participants reported that their older partners had more sexual experience, contributing to an uneven power dynamic in the relationship. Below, a twenty-two-year-old Hispanic college student described how she feared being perceived negatively by her older partner if she inquired about condom use.

*He was a little bit older than me, and I guess I was shy just because I was like, oh, he's more experienced. Let me not ask these questions: what if he looks down on me?*

Trustworthiness was another partner characteristic that participants factored into their STI risk perception analysis and dual-method contraceptive decision-making. For example, a twenty-year-old Hispanic college student explained how trust plays a vital role in her dual-method contraceptive use decision-making.

*I personally don't really use condoms because the people who are my partners are usually people that I trust.*

Meanwhile, several participants attributed a prior STI diagnosis to a partner that they perceived as trustworthy. A nineteen-year-old college student who identified as African American, Hispanic, and White provided insight into a prior STI diagnosis:

*The first time I ever had condomless sex, I trusted the person because they said they get tested regularly, and that was the first and last time I contracted an STI, which was chlamydia.*

Participants also made assumptions about their partner's STI status based on their partner's sexual history as participants were less likely to use condoms with partners whom they believed to have had a low number of sexual partners. In the quote below, a twenty-year-old Hispanic college student assumed her partner's STI status based on his sexual history:

*Because I feel like he hasn't been with many. At the top of my head probably like two to three before me, so it wasn't like, oh my god he's been with more than ten. I should probably have him get checked out before or anything like that.*

### Relationship dynamic

Monogamy, actual or perceived, was a major factor AYACF considered with dual-method contraceptive decision-making as participants reported less condom use while being in a perceived monogamous relationship compared to those who were having casual sex. Participants made assumptions about their partner's STI status based on their partners being perceived as *faithful* or *loyal.* Below, a twenty-year-old African American college student reflected on her thought process in a previous relationship.

*I'm not gonna wear no condom with this person because he's probably never gonna cheat on me or leave me, you know?*

Participants reported that they were more likely to wear condoms while in a monogamous relationship than in a non-monogamous relationship. For example, a twenty-year-old African American participant with a high school education said, *I just didn't use them* [condoms] *as much because I was in a relationship*.

In some instances, perceived monogamy played a role in participants' decision to defer to their partner for condom use decision-making. For example, an eighteen-year-old Hispanic participant with a high school education reported:

*He started saying that we were already together, so what was the reason for us to, you know, use condoms if we were just with each other? But I mean, he did cheat on me, so, because obviously I got chlamydia.*

Additionally, familiarity played an important role in determining condom use. For example, condom use was low in relationships where participants reportedly knew their partner for a period of time prior to engaging in sexual intercourse with them. In the quote below, a twenty-four-year-old African American college student provided insight into how familiarity played a role in her dual-method contraceptive decision-making.

*Well, even before then, the person that gave me the last one* [STI], *we used to date back in middle school. So, I felt comfortable with him, you know, it wasn't like some dude off the street.*

### STI testing as reassurance

Frequent STI testing was common among participants who did not practice consistent condom use. Participants often assumed that STI testing (without condom use) was enough to protect them from STIs, Participants reported that STI testing provided them with *reassurance* after engaging in sexual intercourse without a condom. A twenty-four-year-old, Hispanic college graduate discussed her experience:

*I would just go ahead and book an appointment literally for that week and tell the doctor what was going on, just because I'm very paranoid to some extent, like because I knew I was, like, having casual sex [without a condom].*

Additionally, some participants decided to completely stop wearing condoms because they were getting tested frequently for STIs. In the quote below, a twenty-three-year-old, Hispanic high school graduate with a prior STI diagnosis explained further:

*Yeah, I'm way more comfortable not using condoms anymore. Yeah, but again, I'm emphasizing like I'm emphasizing getting a screen or getting tested with my partner or making sure that they've been tested recently and then if I do just have sex with them, I make sure I'm like, okay in three months, I'm gonna get tested and I'll see.*

Among some participants who continued to have sex with partners who had previously given them an STI, STI testing was preferred to using condoms. A twenty-four-year-old African American college student shared her experience:

*Even when, like, when I found out my spouse was cheating, every time we had sex, I set a doctor's appointment, I don't care if it was every week, every two weeks, every three weeks, every month, I always set a doctor's appointment afterwards.*

**STI diagnosis: A new reality**

For many participants, an STI diagnosis created a shift in STI risk perception analysis. As participants grappled with their STI diagnosis, they began to consider implementing behavioral changes to reduce their likelihood of acquiring another STI in the future. Below, a twenty-three-year-old Hispanic participant with a high school education explained:

*And that kind of put me in the spot where it's kinda like, oh, I should start watching, you know, watching out who I mess with and who I, I should use more protection.*

Similarly, a twenty-year-old Hispanic, college student shared:

*So, I was like, in my mind, like, biological, thinking like I can't get pregnant on my period so I didn't really put too much effort into like using a condom, but after chlamydia, I'm like use a condom every time like it's not negotiable.*

Following an STI diagnosis, some participants reported behavior change as they stopped deferring to their partners for condom use decision-making and started demanding condom use from their sexual partners. A twenty-two-year-old Hispanic college student shared her experience:

*I kind of did put my foot down; I was like, look, it's either we're gonna wear condoms or we're just gonna leave it out here, I was like, I enjoy being with you, obviously, but if you are not gonna follow through with this, then there's no point of us being like that.*

As participants began to take control of their health, they reported experiencing a newfound sense of confidence. For example, a twenty-year-old African American high school graduate said, *When you put your foot down, it feels empowering*. Similarly, a twenty-two-year-old Hispanic college student shared:

*But you demanding a condom, it's like you know what you want in your life, you take care of yourself, you're aware of what can happen and consequences when having sex.*

## Discussion

This grounded theory study provides insight into the complex decision-making processes considered with STI risk perception analysis and dual-method contraceptive use among AYACF in Texas, a population who has experienced an increase

in STI rates over the last five years [1]. We found that AYACF are making assumptions about the STI status of their sexual partners and deferring to them for condom use decision-making. Rather than inquire about their partner's STI status or ask them to wear a condom, participants conducted their own risk perception analysis based on partner characteristics and relationship dynamics.

Specifically, perceived trustworthiness and monogamy were the most cited reasons for not practicing dual-method contraceptive use and deferring to partners for condom use decision-making. Further, among participants with a prior STI diagnosis, perceiving their partner as trustworthy and monogamous were the most common reasons for not using a condom during the sexual encounter that led to their STI diagnosis. Our findings coincide with previous studies that found that young adult women who reported high levels of trust and perceived monogamy with a sexual partner were less likely to use a condom during sexual intercourse than those with low levels of trust and perceived monogamy [54,55].

A possible explanation for the relationship between perceived trustworthiness and monogamy and partner deferral is that young adult women are more likely to be attached to societal romantic love ideals such as love and faithfulness in their relationship than their male partners, leading them to misjudge trust [56]. Additionally, partner deferral is more likely to occur in relationships with an uneven power dynamic than in relationships where decision-making and power are equally distributed [57]. Such findings were reflected in our study as participants who reported having older and more experienced sexual partners were more likely to defer to them for condom use decision-making.

Patriarchal and societal and cultural norms and expectations is a prevalent and complex issue in the US, and can have devastating emotional, physical, and psychological consequences on girls and young women [58,59]. Future research should focus on developing interventions that help AYACF view males partners from a realistic, as opposed to idealistic, perspective, and increase self-esteem. Concurrently, health care providers can educate young women on the likelihood and reality of acquiring an STI or HIV in relationships where partners are perceived as trustworthy or monogamous.

An important finding in this study was the role mothers played in STI risk perception and dual-method contraceptive decision-making. Participants with mothers who avoided sexual health discussions were more likely to have been diagnosed with an STI than participants whose mothers engaged in conversations about sexual health-related topics such as STIs and pregnancy prevention. Further, participants whose mothers personally experienced or had a close family member experience unintended pregnancy as an adolescent or young adult were more likely to have received information regarding STI and pregnancy prevention from their mothors compared to participants with mothers who had not experienced an unintended pregnancy at a young age.

Mother-daughter communication has been found to reduce STI and unintended pregnancy among adolescents under 18 years of age [60–62]. Our findings demonstrate that mother-daughter communication in adolescence may also reduce STI and unintended pregnancy risk in young adulthood. Considering the powerful role mothers played in STI and pregnancy prevention, future research and practice should focus on how to promote and facilitate sexual health communication between mothers and daughters. Specifically, research is needed to develop, implement, and evaluate the effectiveness of mother-daughter communication models and workshops. Additionally, pediatric or gynecological health care appointments can be an opportunity for health care providers to facilitate conversations about contraeption or STIs between females under the age of 18 and their mothers or maternal guardians who accompany them.

Another finding from this study is that participants used STI testing as reassurance after engaging in sexual intercourse without a condom. Conversely, a previous study found that young adult women were using STI testing as a form of reassurance before engaging in sexual intercourse without a condom [11]. One possible explanation why this study's participants using STI testing as reassurance is that they were recruited from clinics with free sexual health services (including STI testing), eliminating any potential financial barriers to STI testing.

Another possible explanation for participants using STI testing for reassurance after engaging in sexual intercourse without a condom is that participants perceive STIs as temporary illnesses that are treatable or curable, and aren't knowledgable or overly concerned about lifelong diseases such as HIV, HPV or HSV. Further, frequent STI testing might be

used as a substitue for engaging in condom negotiation with partners prior to sex, a task that could be viewed as uncomfortable or confrontational. While research has demonstrated that increased access to free or affordable health care results in disease reduction [63–65], there is limited understanding of how having access to free health care influences sexual health decision-making in the United States, a country where the majority of the population do not have access to government-provided health care [66]. Additionally, our findings highlight the importance of providers educating patients that STI testing is not a substitute for condom use and that some HIV and STIs such as as HPV and HSV are lifelong diseases.

Additionally, societal factors such as state abortion policy and school-based sex education may have played a significant role in perpetuating fear of unplanned pregnancy among participants. Our study found that recent changes to abortion laws likely exacerbated participants' pre-existing fear of pregnancy as participants reported buying pregnancy tests and Plan B in bulk, as well as considering potentially life-threatening measures to self-induce an abortion after Roe V. Wade was overturned. Meanwhile, almost all participants reported having a sex education that focused on abstinence and avoiding pregnancy but did not provide information about STIs or STI prevention. Thus, their reduced fear of STIs in comparison to pregnancy might be attributed to a lack of knowledge about STIs and their health-related consequences. While it is is known that states with conservative sex education policies have some of the highest STI and abortion rates (prior to June 2022) [67,68], the relationship between the specific type of sex education curricula provided in public schools (i.e., abstinence-only, abstinence-plus, comprehensive…) and abortion needs to be explored further. Further, our findings highlight the need to reform sex education policy to implement sex education curricula that provides young girls with the knowledge and tools needed to reduce their risk of STIs and HIV and engage in healthy relationships. The current lack of evidence-based and comprehensive sex education provided to young girls presents an opportunity for future research to focus on interventions that can assist healthcare providers in filling knowledge gaps.

## Limitations

This study has limitations. First, most participants were in college and identified as Black/African American and/or Hispanic, and therefore the findings may not be applicable to AYACF who do not attend college and/or belong to other racial or ethnic groups. Second, all participants had access to free STI testing and treatment. Thus, findings might not be transferable to populations who do not have free access to contraception nor sexual health care and services. Further, access to free sexual health care might have influenced participants' decision to engage in dual-method contraceptive use. Third, this study was cross-sectional, limiting the understanding of the participants' behavior over a period of time. Fourth, all interviews were conducted virtually via Zoom, which could have limited the researcher's ability to interpret participants' overall body language since participants were only viewed from above the waist. Fifth, self-reporting bias could have occurred, as participants potentially felt a need to provide socially "correct" answers pertaining to their sexual health practices (i.e., consistent condom use and conversations with partners about STIs and condom use). Additionally, participants might have felt embarrassed to truthfully share their sexual health practices due to the nature of the subject. Recall bias was, also, a risk as some participants reported on sexual counters that occurred more than three years prior to the interview. Furthermore, the interview guide developed and implemented by the research team was not tested prior to being used in this study. Testing the interview guide prior to study implementation could have validated questions in the interview guide or lead to the development of new questions that more accurately capture factors associated with dual-method contraceptive decision-making or STI risk perception. Another limitation is that the perspective of male partners was not included in this study. Future research should examine the perspectives of both male and female sexual partners to provide a more in-depth and well-rounded understanding of condom use decision-making. Last, findings regarding the role of abortion laws and sex education were self-reported, reflecting participants perception of how they influenced contraceptive decision-making.

## Conclusion

Our results indicate that dual-method contraceptive decision-making and STI risk perception analysis is a complex process involving personal, relationship, and societal factors. AYACF likely have low STI risk perception, are not practicing dual-method contraceptive use, and are deferring condom use decision-making to partners they perceive as trustworthy or monogamous. Once diagnosed with an STI, AYACF likely stop relying on perceptions and assumptions to determine the need for dual-method contraceptive use and begin to engage in new thinking and behaviors. The shift in perspective and behavior caused by the STI diagnosis highlights the need to empower young women to put their own sexual health needs before their partners.

Additionally, AYACF have misconceptions about the severity and prevalence of STI's and HIV and the role of STI and HIV testing. Among AYACF prescribed contraceptive users, the fear of pregnancy was far greater than the risks of STIs and HIV contributing to minimal and/or infrequent condom use. Rather than practice dual-method contraceptive use, AYACF with affordable and accessible STI testing, got tested for STI's after engaging in sexual intercourse without a condom. Mother-daughter communication and comprehensive sex education can help correct these misconceptions and may increase dual-method contraceptive use, reducing STI and HIV risk. Future research should focus on interventions to promote and support mother-daughter sexual health communication and comprehensive sex education in the AYACF population.

## Author contributions

**Conceptualization:** Lauren Holt, Amy Corneli, Eleanor Stevenson, Jonas J. Swartz, Michael V. Relf.

**Data curation:** Lauren Holt.

**Formal analysis:** Lauren Holt, Michael V. Relf.

**Funding acquisition:** Lauren Holt.

**Investigation:** Lauren Holt.

**Methodology:** Lauren Holt, Amy Corneli, Michael V. Relf.

**Supervision:** Michael V. Relf.

**Validation:** Michael V. Relf.

**Visualization:** Lauren Holt, Michael V. Relf.

**Writing – original draft:** Lauren Holt.

**Writing – review & editing:** Lauren Holt, Amy Corneli, Eleanor Stevenson, Jonas J. Swartz, Michael V. Relf.

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
