## [Decision Letter · Decision Letter 0]

14 Feb 2025

PONE-D-24-49168
The Theoretical Understanding of Risk Perception and Dual-Method Contraceptive Decision-Making among Texas Adolescent and Young Adult Cisgender Females
PLOS ONE

Dear Dr. Holt,

Thank you for submitting your manuscript to PLOS ONE. After careful consideration, we feel that it has merit but does not fully meet PLOS ONE’s publication criteria as it currently stands. Therefore, we invite you to submit a revised version of the manuscript that addresses the points raised during the review process.

We look forward to receiving your revised manuscript.

Kind regards,

Obasanjo Afolabi Bolarinwa

Academic Editor

PLOS ONE

Journal Requirements:

3. We note that your Data Availability Statement is currently as follows: All relevant data are within the manuscript and its Supporting Information files

6. Please ensure that you refer to Figure 1 in your text as, if accepted, production will need this reference to link the reader to the figure.

Reviewers' comments:

Reviewer's Responses to Questions

**Comments to the Author**

1. Is the manuscript technically sound, and do the data support the conclusions?

Reviewer #1: Yes

Reviewer #2: Partly

2. Has the statistical analysis been performed appropriately and rigorously?

Reviewer #1: N/A

Reviewer #2: N/A

3. Have the authors made all data underlying the findings in their manuscript fully available?

Reviewer #1: Yes

Reviewer #2: No

4. Is the manuscript presented in an intelligible fashion and written in standard English?

Reviewer #1: Yes

Reviewer #2: Yes

5. Review Comments to the Author

Reviewer #1: Structure and Organization

•The introduction section is quite lengthy and could be condensed to focus more directly on the study objectives and rationale. Consider moving some of the background information on STI rates and contraceptive use to a separate background section.

•The methods section would benefit from clearer subheadings to delineate the different components (e.g., Study Design, Participants, Data Collection, Data Analysis).

•Consider adding a separate "Theoretical Framework" section to explain the grounded theory approach in more detail, as this is central to the study methodology.

Content and Clarity

•The research questions/aims could be stated more explicitly early in the introduction. Currently they are embedded in the "Study Purpose" subsection.

•Provide more details on the data analysis procedures used, including the specific coding approach and how themes were developed.

•Include information on how many participants were recruited through each method (clinic vs. referral).

•Clarify whether data saturation was reached and how this was determined.

Methodological Considerations

•Explain the rationale for conducting interviews virtually rather than in-person and discuss any potential limitations of this approach.

•Provide more information on how trustworthiness and rigor were ensured in the data collection and analysis process.

•Consider discussing potential limitations of the single-coder approach to data analysis.

Discussion and Implications

•Expand the discussion of how the findings relate to existing literature and theory on contraceptive decision-making.

•Provide more specific recommendations for practice and future research based on the study findings.

•Discuss the transferability of the findings to other contexts or populations.

Reviewer #2: The background would benefit from reducing redundancy, clarifying its focus, integrating sections more cohesively, and providing clear research objectives. Addressing these weaknesses will enhance its impact and readability. The methods need more detail to ensure replicability of the study. Some information is missing from the results section. Please revise the discussion section.

Please see an uploaded document for comments

6. PLOS authors have the option to publish the peer review history of their article (what does this mean?). If published, this will include your full peer review and any attached files.

Reviewer #1: **Yes:** Dr Chris Visser

Reviewer #2: No

---

## [Author Response · Author response to Decision Letter 1]

12 Jun 2025

Dear editors of PLOS ONE,

Thank you for the opportunity to revise our manuscript, “The Theoretical Understanding of Risk Perception and Dual-Method Contraceptive Decision-Making among Texas Adolescent and Young Adult Women” (PONE-D-24-49168).

We have provided responses to each of the questions from the two reviewers below.

We would be happy to provide any additional information that may be needed.

---

## [Decision Letter · Decision Letter 1]

10 Sep 2025

PONE-D-24-49168R1
The Theoretical Understanding of Risk Perception and Dual-Method Contraceptive Decision-Making among Texas Adolescent and Young Adult Cisgender Females
PLOS ONE

Dear Dr. Holt,

Thank you for submitting your manuscript to PLOS ONE. After careful consideration, we feel that it has merit but does not fully meet PLOS ONE’s publication criteria as it currently stands. Therefore, we invite you to submit a revised version of the manuscript that addresses the points raised during the review process.

We look forward to receiving your revised manuscript.

Kind regards,

Shadab Shahali, PHD

Academic Editor

PLOS ONE

Journal Requirements:

Additional Editor Comments:

Reviewer #1:

This study could potentially make an important contribution to understanding contraceptive and STI prevention decision-making among a high-risk, yet underexplored group. The perspectives from participants are likely to be valuable to researchers, clinicians, and public health policymakers.

The impact of abortion laws and sex education is inferred from self-reports and may not necessarily capture broader societal trends. The authors should clarify that these are perceptions rather than measured behavioral outcomes.

The description of coding lacks detail about how the two analysts resolved discrepancies and whether coding decisions were audited or validated beyond mutual discussion.

A further suggestion would be that since maternal influences and partner dynamics emerged as critical elements in the study's findings, the authors could consider highlighting these as recommended avenues for intervention, for example, mother-daughter workshops or developing partner communication models. The authors could also consider discussing policy and clinical implications more explicitly (e.g., the need for sexual health education curricular reforms, or the development of targeted strategies for counseling)

Reviewer #3:

Dear author,

Your work subject is very important and based on the provided information, it shows still there are a lot of problem to improve knowledge, attitude, practice and behavior among relevant target groups.

I think your manuscript needs some improvement and I provided my comments on the file.

Please pass through them.

Hope you find them helpful

Reviewers' comments:

Reviewer's Responses to Questions

**Comments to the Author**

1. If the authors have adequately addressed your comments raised in a previous round of review and you feel that this manuscript is now acceptable for publication, you may indicate that here to bypass the “Comments to the Author” section, enter your conflict of interest statement in the “Confidential to Editor” section, and submit your "Accept" recommendation.

Reviewer #1: All comments have been addressed

Reviewer #3: (No Response)

2. Is the manuscript technically sound, and do the data support the conclusions?

Reviewer #1: Partly

Reviewer #3: Partly

3. Has the statistical analysis been performed appropriately and rigorously?

Reviewer #1: Yes

Reviewer #3: N/A

4. Have the authors made all data underlying the findings in their manuscript fully available?

Reviewer #1: No

Reviewer #3: Yes

5. Is the manuscript presented in an intelligible fashion and written in standard English?

Reviewer #1: Yes

Reviewer #3: No

6. Review Comments to the Author

Reviewer #1: This study could potentially make an important contribution to understanding contraceptive and STI prevention decision-making among a high-risk, yet underexplored group. The perspectives from participants are likely to be valuable to researchers, clinicians, and public health policymakers.

The impact of abortion laws and sex education is inferred from self-reports and may not necessarily capture broader societal trends. The authors should clarify that these are perceptions rather than measured behavioral outcomes.

The description of coding lacks detail about how the two analysts resolved discrepancies and whether coding decisions were audited or validated beyond mutual discussion.

A further suggestion would be that since maternal influences and partner dynamics emerged as critical elements in the study's findings, the authors could consider highlighting these as recommended avenues for intervention, for example, mother-daughter workshops or developing partner communication models. The authors could also consider discussing policy and clinical implications more explicitly (e.g., the need for sexual health education curricular reforms, or the development of targeted strategies for counseling)

Reviewer #3: Dear author,

Your work subject is very important and based on the provided information, it shows still there are a lot of problem to improve knowledge, attitude, practice and behavior among relevant target groups.

I think your manuscript needs some improvement and I provided my comments on the file.

Please pass through them.

Hope you find them helpful.

7. PLOS authors have the option to publish the peer review history of their article (what does this mean?). If published, this will include your full peer review and any attached files.

Reviewer #1: **Yes:** Dr Christiaan Visser

Reviewer #3: **Yes:** Mohammad Eslami

---

## [Author Response · Author response to Decision Letter 2]

1 Dec 2025

Please see Response to Reviewers attachment.

---

## [Decision Letter · Decision Letter 2]

23 Dec 2025

PONE-D-24-49168R2
The Theoretical Understanding of Risk Perception and Dual-Method Contraceptive Decision-Making among Texas Adolescent and Young Adult Cisgender Females
PLOS One

Dear Dr.Lauren Holt,

Thank you for submitting your manuscript to PLOS ONE. After careful consideration, we feel that it has merit but does not fully meet PLOS ONE’s publication criteria as it currently stands. Therefore, we invite you to submit a revised version of the manuscript that addresses the points raised during the review process.

We look forward to receiving your revised manuscript.

Kind regards,

Shadab Shahali, PHD

Academic Editor

PLOS One

Journal Requirements:

Reviewers' comments:

Reviewer's Responses to Questions

**Comments to the Author**

1. If the authors have adequately addressed your comments raised in a previous round of review and you feel that this manuscript is now acceptable for publication, you may indicate that here to bypass the “Comments to the Author” section, enter your conflict of interest statement in the “Confidential to Editor” section, and submit your "Accept" recommendation.

Reviewer #1: All comments have been addressed

Reviewer #3: (No Response)

2. Is the manuscript technically sound, and do the data support the conclusions?

Reviewer #1: Yes

Reviewer #3: Yes

3. Has the statistical analysis been performed appropriately and rigorously?

Reviewer #1: N/A

Reviewer #3: I Don't Know

4. Have the authors made all data underlying the findings in their manuscript fully available?

Reviewer #1: No

Reviewer #3: Yes

5. Is the manuscript presented in an intelligible fashion and written in standard English?

Reviewer #1: Yes

Reviewer #3: Yes

6. Review Comments to the Author

Reviewer #1: The authors have been responsive to the reviewers’ feedback, and, in its current form, the manuscript is broadly suitable for publication pending only minor editorial polishing by the journal (mainly style and copy-editing). Minor issues that the journal editors can readily address and which do not warrant another round of peer review include occasional typographical spacing or duplicated words (e.g., “effective effective” on page 3).

Reviewer #3: Thanks for addressing some of the comments.

There are still some comments which need to be addressed.

Looking at condom as a dual protection method should be considered throughout of the text. It is not satisfactory to consider it in a limited part of the paper.

Please have a second review on all comments and address the remained.

7. PLOS authors have the option to publish the peer review history of their article (what does this mean?). If published, this will include your full peer review and any attached files.

Reviewer #1: **Yes:** Dr Christiaan Visser

Reviewer #3: **Yes:** Mohammad Eslami

---

## [Author Response · Author response to Decision Letter 3]

6 Feb 2026

Dear editors of PLOS ONE,

Thank you for the opportunity to revise our manuscript, “The Theoretical Understanding of Risk Perception and Dual-Method Contraceptive Decision-Making among Texas Adolescent and Young Adult Women” (PONE-D-24-49168).

We have provided responses to each of the questions from the two reviewers in the Response to Reviewers attachment.

We are happy to provide any additional information that may be needed.

Sincerely,

Lauren Holt

---

## [Editor Report · Decision Letter 3]

11 Feb 2026

The Theoretical Understanding of Risk Perception and Dual-Method Contraceptive Decision-Making among Texas Adolescent and Young Adult Cisgender Females

PONE-D-24-49168R3

Dear Dr. Holt,

We’re pleased to inform you that your manuscript has been judged scientifically suitable for publication and will be formally accepted for publication once it meets all outstanding technical requirements.

Kind regards,

Shadab Shahali, PHD

Academic Editor

PLOS One
---

## [Editor Report · Acceptance letter]

PONE-D-24-49168R3

PLOS One

Dear Dr. Holt,

I'm pleased to inform you that your manuscript has been deemed suitable for publication in PLOS One. Congratulations! Your manuscript is now being handed over to our production team.

Kind regards,

on behalf of

Dr. Shadab Shahali

Academic Editor

PLOS One